# RESOLVE: RELATIONAL REASONING WITH SYMBOLIC AND OBJECT-LEVEL FEATURES USING VECTOR SYMBOLIC PROCESSING

## ABSTRACT

Modern transformer-based encoder-decoder architectures struggle with reasoning tasks due to their inability to effectively extract relational information between input objects (data/tokens). Recent work introduced the *Abstractor* module, embedded between transformer layers, to address this gap. However, the Abstractor layer while excelling at capturing relational information (pure relational reasoning), faces challenges in tasks that require both object and relational-level reasoning (partial relational reasoning). To address this, we propose RESOLVE, a neuro-vector symbolic architecture that combines object-level features with relational representations in high-dimensional spaces, using fast and efficient operations such as bundling (summation) and binding (Hadamard product) allowing both object-level features and relational representations to coexist within the same structure without interfering with one another. RESOLVE is driven by a novel attention mechanism that operates in a bipolar high dimensional space, allowing fast attention score computation compared to the state-of-the-art. By leveraging this design, the model achieves both low compute latency and memory efficiency. RESOLVE also offers better generalizability while achieving higher accuracy in purely relational reasoning tasks such as sorting as well as partial relational reasoning tasks such as math problem-solving compared to state-of-the-art methods.

## 1 INTRODUCTION

Analogical reasoning, which involves recognizing abstract relationships between objects, is fundamental to human abstraction and thought. This contrasts with semantic (meaning-based) and procedural (task-based) knowledge acquired from sensory information, which is typically processed through contemporary approaches like deep neural networks (DNNs). However, most of these techniques fail to extract abstract rules from limited samples Barrett et al. (2018); Ricci et al. (2018); Lake & Baroni (2018).

These reasoning tasks can be *purely* or *partially* relational. Figure 1 presents an example of a purely relational task where the objects (e.g. frog, mountains) are *randomly* generated. In this task, only the information representing relationships *between* the objects is relevant, not the objects themselves. By contrast, in Figure 2a the purpose is to learn the abstract rule of subtraction, which is unknown to the model, from pairs of MNIST digits. This abstract rule relies on the *relational representation* between the digits (derived from their relationship with one another, in this case their ordering) and the digits themselves (the

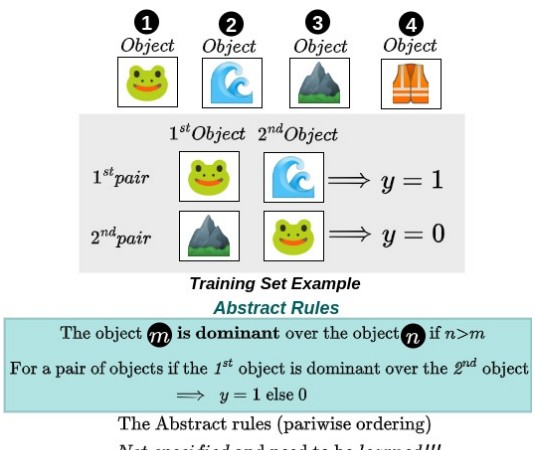

Figure 1: Example of purely relational task: *Pairwise Ordering*

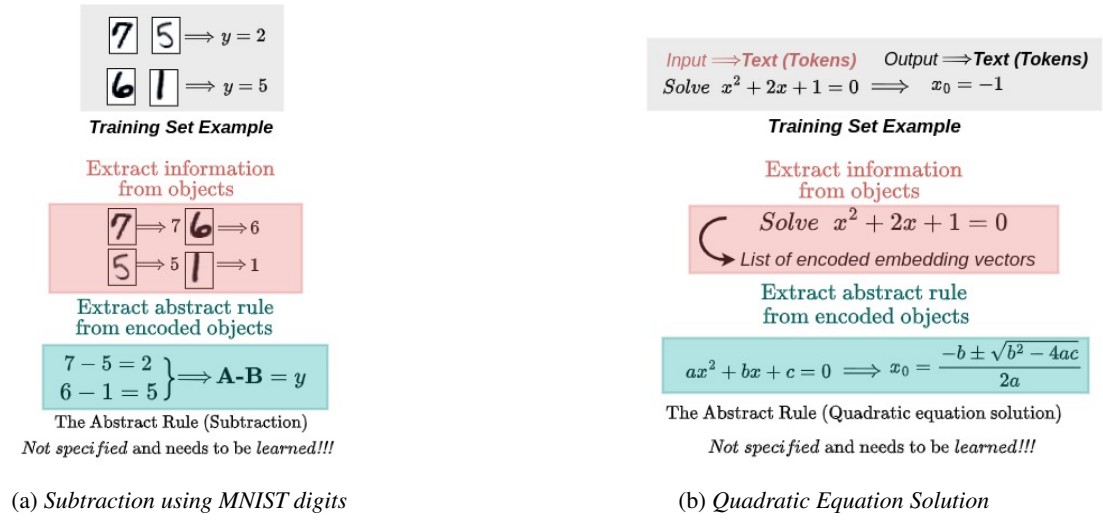

(a) *Subtraction using MNIST digits*  (b) *Quadratic Equation Solution*

Figure 2: Two examples of partially relational tasks(Figure 2a and 2b)

values being subtracted), which are *object* features. This is a *partially* relational problem. Similarly, in Figure 2b, the purpose is to learn the abstract rule of the quadratic formula (i.e. the solution to the quadratic problem shown at the bottom of Figure 2b) from the object features (derived from the text tokens representing equation coefficients) and the relational representation (derived from the coefficient ordering).

These relational or partially relational tasks have been shown to be problematic for transformer-based architectures (Altabaa et al., 2023), which encode both the object features and relational representations into the same structure. (Altabaa et al., 2023) instead created a learnable inductive bias derived from the transformer architecture for explicit relational reasoning. Although this solution is sufficient for purely relational tasks such as Figure 1, it is less efficient for partially relational tasks such as Figures 2a and 2b where the object features and relational representations are both significant.

The poor ability of transformers to superpose relational representations and object-level features is due to the low dimensionality of their components, causing interference between object features and relational representations (Webb et al., 2024b). By contrast, vector symbolic architectures (VSA) have used high-dimensional spaces to superpose object features and relational representations with low interference Hersche et al. (2023). Transformer-based architectures are moreover known to be power-inefficient due to the attention score computation Debus et al. (2023). Vector symbolic architectures have been proven to be power-efficient Menet et al. (2024) with low memory overhead due to the low-bitwidth (bipolar) representation of high-dimensional vectors. However, current VSA techniques require prior knowledge of abstract rules and a pre-engineered set of relations and objects (e.g., blue, triangle), making them unsuitable for sequence-to-sequence reasoning.

These arguments motivate the design of RESOLVE, an innovative vector symbolic architecture allowing superposition of relational representations and object-level features in high dimensional spaces. Object-level features are encoded through a novel, fast, and efficient *HD-attention* mechanism. The key contributions of this paper are:

- We are the *first* to propose a strategy for addressing the relational bottleneck problem (capturing relational information between data/objects rather than input data/object attributes or features from limited training data) using a *vector symbolic architecture*. Our method captures relational representations of input objects in a hyperdimensional vector space, while maintaining object-level information and features in the *same representation structure*, while minimizing their interference with each other at the same time. The method outperforms prior art in tasks that require both pure and partial relational reasoning.

- We implement a novel, fast and efficient attention mechanism that operates directly in a bipolar ($\{-1, 1\}$) high-dimensional space. Vectors representing relationships between

symbols are learned, eliminating the need for prior knowledge of abstract rules required by prior work Hersche et al. (2023).s

- Our system significantly reduces computational costs by simplifying attention score matrix multiplication to bipolar operations and relying on lightweight high-dimensional operations such as the Hadamard product (also known as *binding*).

In the following section we discuss related prior work, followed by an overview of our symbolic HD-attention mechanism in Section 3. We then discuss our vector-symbolic hyperdimensional attention mechanism and contrast it to the relational bottleneck approach in Section 4. The `RESOLVE` encoder and hypervector bundling is then discussed in Section 5 and the full architecture in Section 6. We then present experimental validation in Section 7, followed by conclusions.

## 2 RELATED WORK

To address the problem of learning abstract rules, *symbolic AI* architectures such as the Relation Network (Santoro et al., 2017) propose a model for pairwise relations by applying a Multilayer Perceptron (MLP) to concatenated object features. Another example, PrediNet (Shanahan et al., 2020), utilizes predicate logic to represent relational features. Symbolic AI approaches combined with neural networks were leveraged by *neurosymbolic learning* Manhaeve et al. (2018); Badreddine et al. (2022); Barbiero et al. (2023); Xu et al. (2018) to improve this rule learning, with optimizations such as logical loss functions Xu et al. (2018) and logical reasoning networks applied to the predictions of a neural network Manhaeve et al. (2018). However, these systems require prior knowledge of the abstract rules guiding a task. They also require pre-implementation of object attributes Hersche et al. (2023) (e.g., red, blue, triangle, etc.). This approach is only feasible for simple tasks (e.g., Raven's Progressive Matrices Raven (1938)) and is not appropriate for complex sequence-based partially relational tasks such as the quadratic solution of Figure 2b.

For sequence-based partially relational tasks such as the math problem-solving of Figures 2a and 2b an *encoder-decoder structure with transformers* has been used Saxton et al. (2019). However, transformers often fail to capture explicit relational representations.

A solution to the shortcomings of encoder-decoder approaches is proposed in (Webb et al., 2024b), using the *relational bottleneck* concept. This aims to separate relational representations learned using a learnable inductive bias from object-level features learned using connectionist encoder-decoder transformer architectures or DNNs. Several models are based on this idea: *CoRelNet*, introduced in (Kerg et al., 2022), simplifies relational learning by modeling a similarity matrix. A recent study (Webb et al., 2020) introduced an architecture inspired by Neural Turing Machines (NTM) (Graves et al., 2014), which separates relational representations from object features. Building on this concept, the Abstractor (Altabaa et al., 2023) adapted Transformers (Vaswani et al., 2017) for abstract reasoning tasks by creating an 'abstractor'—a mechanism based on cross-attention applied to relational representations for sequence-to-sequence relational tasks. A model known as the Visual Object Centering Relational Abstract architecture (OCRA) (Webb et al., 2024a) maps visual objects to vector embeddings, followed by a transformer network for solving symbolic reasoning problems such as Raven's Progressive Matrices (Raven, 1938). A subsequent study (Mondal et al., 2024) combined and refined OCRA and the Abstractor to address similar challenges. However, these relational bottleneck structures still suffer from the drawback of intereference between the relational representations and object features in deep layers due to their lower feature dimensionality (Webb et al., 2024b).

However, recent work (Hersche et al., 2023) has shown that *Vector Symbolic Architectures (VSAs)*, a neuro-symbolic paradigm using high-dimensional vectors (Kanerva, 2009) with a set of predefined operations (e.g., element-wise addition and multiplication), exhibit strong robustness to vector superposition as an alternative to the relational bottleneck. In addition, Hyperdimensional Computing (HDC) is recognized for its low computational overhead (Mejri et al., 2024b; Amrouch et al., 2022) compared to transformer-based approaches. However, prior work on VSAs has relied on pre-engineered set of objects and relations, limiting their applicability to sequence-to-sequence reasoning tasks.

In contrast to prior research, this paper is the first to leverage VSAs to efficiently combine object-level information with relational information in high-dimensional spaces, taking advantage of the lower

interference between object features and relational representations in high dimensions. We also propose the first efficient attention mechanism for high-dimensional vectors (*HD-Attention*).

# 3 OVERVIEW

In this section we present an overview of prior architectures used to learn abstract rules, and use them to illustrate the unique features of our VSA-based architecture.

Figure 3a illustrates the self-attention mechanism used in transformer architectures Vaswani et al. (2017). In step $t_1$, objects are first encoded into keys, queries, and values. In step $t_2$, self-attention captures correlations between keys and queries in an *attention score* matrix. Finally, in step $t_3$, this matrix is used to mix values and create encoded outputs. Self-attention is thus designed to capture correlations between input object sequence elements. However, it fails to capture relational representations of the input object sequence Altabaa et al. (2023), leading to poor generalization capability for abstract rule-based tasks. The *Abstractor* mechanism aims to fix that flaw.

Figure 3b illustrates the Abstractor mechanism. In step $a_1$ of the abstractor architecture shown in Fig. 3b, objects are first encoded into queries and keys, which are used to build attention scores (step $a_2$) similar to self-attention. In parallel, in step $a_3$, a set of symbols (learnable inductive biases) consisting of a set of trainable vectors are encoded into values. In step $a_4$, the attention scores and the symbols are used to generate the abstract outputs, a dedicated structure for relational representations Altabaa et al. (2023); Webb et al. (2020) that are disentangled from object-level features. This approach, known as the *relational bottleneck*, separates object-level features from relational representations. However, this separation can make it difficult to learn abstract rules for partially relational tasks.

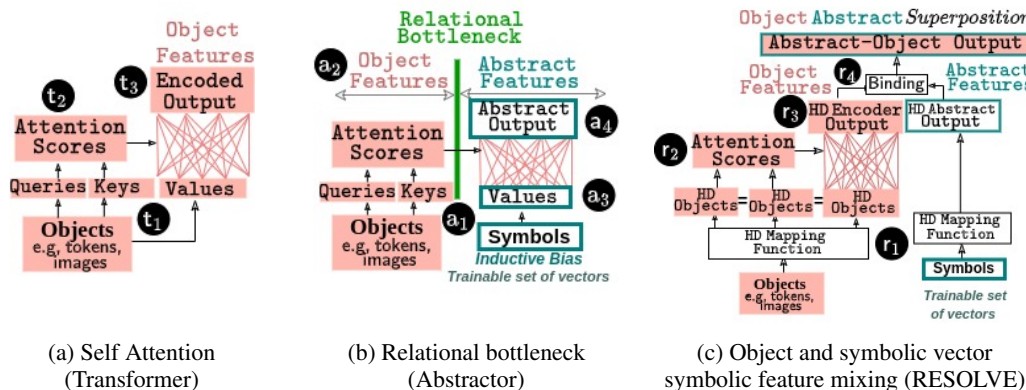

(a) Self Attention (Transformer)    (b) Relational bottleneck (Abstractor)    (c) Object and symbolic vector symbolic feature mixing (RESOLVE)

Figure 3: Comparison of a relational bottleneck approach applied on the transformer (Figure 3b) separating object-level features while keeping only abstract features with a vector symbolic architecture alternative to the relational bottleneck using *binding* to mix object and abstract level information in high dimensional (HD) space with low interference (Figure 3c)

The `RESOLVE` architecture (shown in Figure 3c) explicitly structures the learning of relational information while encoding object-level features. In step $r_1$, objects and symbols are mapped to a high-dimensional (HD) space using an high-dimensional encoder to generate HD Objects (object-level feature representations) and HD Abstract outputs (relational representations). The HD Objects, shown three times, are identical. They are first used in step $r_2$ to compute attention scores. Then, in step $r_3$, these attention scores are used as weights to combine the HD Objects, producing an HD encoded output. In step $r_4$, the HD Abstract output and the HD encoded output are superimposed through a binding operation (Hadamard product) to provide a mixed relational representation and object feature vector in high dimensions, avoiding the interference between relational representations and object features that this mixing causes in lower dimensions (seen in transformers (Webb et al., 2024b)).

## 4 RELATIONAL BOTTLENECK AND VSA APPROACH MODELING

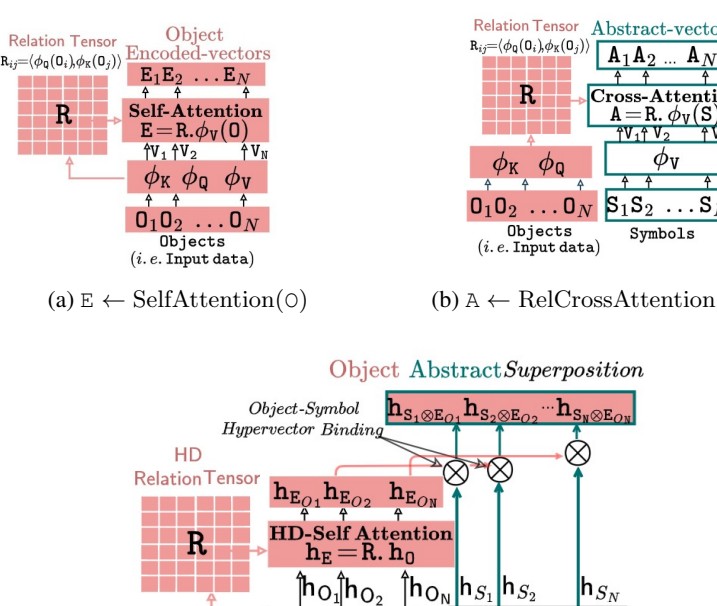

(a) $\texttt{E} \leftarrow \text{SelfAttention}(\texttt{O})$      (b) $\texttt{A} \leftarrow \text{RelCrossAttention}(\texttt{O}, \texttt{S})$

(c) $\texttt{h}_{\texttt{E}_\texttt{O} \otimes \texttt{S}} \leftarrow \text{HD-Attention}(\texttt{h}_\texttt{O}) \otimes \texttt{h}_\texttt{S}$

Figure 4: Comparison between SelfAttention Vaswani et al. (2017) 4a, RelationalCrossAttention Altabaa et al. (2023) 4b and our VSA approach 4c. We show a single head of multi-attention for brevity. The object-related operations are in red and the relational-related (abstract/symbolic) operations are in turquoise.

Figure 4 contrasts the self-attention mechanism (Figure 4a), the relational cross-attention (Figure 4b), and our approach (Figure 4c), using red and turquoise colors. The self-attention mechanism in Figure 4a is applied to a sequence of objects (for instance, token embeddings) denoted by $\texttt{O}_{1..N}$. Each object is of dimension $\texttt{F}$. The objects are encoded into Keys, Queries, and Values through linear projections: $\phi_Q : \texttt{O} \mapsto \texttt{O} \cdot \texttt{W}_Q$, $\phi_K : \texttt{O} \mapsto \texttt{O} \cdot \texttt{W}_K$, and $\phi_V : \texttt{O} \mapsto \texttt{O} \cdot \texttt{W}_V$, where $\texttt{W}_Q, \texttt{W}_K, \texttt{W}_V$ are learnable matrices. The Queries and Keys are used to compute an attention score matrix, which captures the relationships between encoded objects through a pairwise dot product $\langle , \rangle$. (Altabaa et al., 2023) interprets this as a *relation tensor*, denoted by $\texttt{R} = [\langle \phi_Q(\texttt{O}_i), \phi_K(\texttt{O}_j) \rangle] \, i, j = 1^N$. $\texttt{R}$ is normalized to obtain $\overline{\texttt{R}}$ using a $\texttt{Softmax}$ function to produce probabilities. SelfAttention($\texttt{O}$) thus generates a mixed relational representation $\texttt{E}$ of the encoded objects (Values) through the normalized relational tensor $\overline{\texttt{R}}$ (i.e., $\texttt{E}_i = \sum_j; \overline{\texttt{R}}_{ij} \phi_V(\texttt{O}_j)$). A transformer uses the matrix $\overline{\texttt{R}}$ to capture input relations and $\phi_V$ to encode object-level features, but $\phi_V$ is not designed to learn abstract rules.

Figure 4b shows the relational attention mechanism by (Altabaa et al., 2023) that isolates object-level features (in red) from abstract/relational information (in turquoise) to improve abstract rule learning. Like self-attention, the objects $\texttt{O}_{1..N}$ are first encoded into Keys and Queries through the same learnable projection functions $\phi_K$ and $\phi_Q$, which are then used to build a normalized relation tensor $\overline{\texttt{R}}$. In parallel, a set of symbols $\texttt{S}_{1..N}$ ($N$ learnable vectors with the same dimensionality as the objects) are encoded into values using a projection function $\phi_V$ (i.e., $V = \phi_V(\texttt{S})$). The encoded symbols (Values) are mixed using the relation tensor weights through a *relational cross-attention* mechanism to generate a mixed relational representation containing less object-level information and more relational (abstract) information. These are called *Abstract States*, denoted as $\texttt{A}_{1..N}$ ($\texttt{A}_i = \sum_j; \overline{\texttt{R}}_{ij} \phi_V(\texttt{S}_j)$).

Figure 4c shows our VSA-based system. It starts by encoding the objects $\texttt{O}_{1..N}$ from their $\texttt{F}$-dimensional space into a high-dimensional ($\texttt{D}$-dimensional) space using an encoder denoted by $\phi_{\texttt{HD}}$ to generate high-dimensional object vectors $\texttt{h}_{\texttt{O}_{1..N}}$. We extract relational scores from this using a novel *HD-attention mechanism* to build a *HD relation tensor*, denoted as $\texttt{R}$. This matrix is then

normalized through a softmax function to generate $\overline{\mathrm{R}}$. These normalized scores are used to mix the $\mathrm{h}_{\mathrm{O}_{1..N}}$, generating encoded object-level high-dimensional vectors $\mathrm{h}_{\mathrm{E}_{\mathrm{O}_i}} = \sum_j \overline{\mathrm{R}}_{ij}\mathrm{h}_{\mathrm{O}_j}$. A set of learnable symbols $\mathrm{S}_{1..N}$ with the same dimensionality as the objects is used to encode relational information. These symbols are mapped to the high-dimensional space through $\phi_{\mathrm{HD}}$, generating $h_{\mathrm{S}_{1..N}}$. These high-dimensional symbolic vectors are *bound* (i.e., Hadamard product/element-wise multiplication) with the encoded high-dimensional object-level vectors to generate vectors $\mathrm{h}_{\mathrm{E}_{\mathrm{O}\otimes\mathrm{S}}}$ that carry both object-level and relational (abstract) information.

# 5 RESOLVE: HD-ENCODER AND HD-ATTENTION MECHANISM WITH HYPERVECTOR BUNDLING

Figure 5 shows the HD-Encoder and HD-Attention mechanism applied to an input sequence of objects $\mathrm{O}_1$, ..., $\mathrm{O}_{\mathrm{N}}$ (see Figure 4). In Step ❶, objects are mapped from the $F$-dimensional feature space to a $D$-dimensional HD space ($D \sim 10^3$) using the HD encoder $\phi_{\mathrm{HD}}$. We have implemented $\phi_{\mathrm{HD}}$ using single-dimension convolution operations inspired by Mejri et al. (2024a). In this encoder scheme, each object $\mathrm{O}_i$ is convolved with a learnable high dimensional vector called the *HD-Basis* denoted

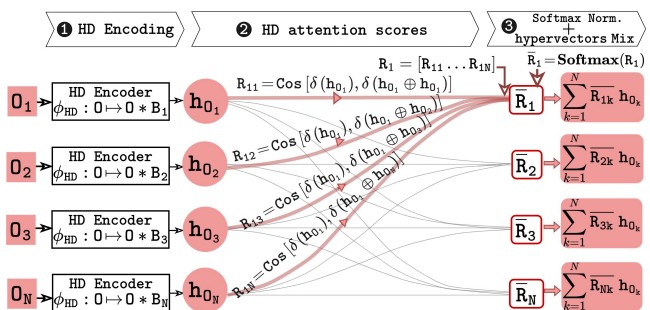

Figure 5: HD-Encoder $\phi_{\mathrm{HD}}$ and HD-Attention($\mathrm{O}_{1...\mathrm{N}}$)

by $\mathrm{B}_i \in \mathbb{R}^{N \times D-F+1}$, giving rise to the high-dimensional HD Object hypervectors $\mathrm{h}_{\mathrm{O}_i}[j] = \sum_k \mathrm{O}_i[k] \cdot \mathrm{B}_i[j-k]$.

Step ❷ consists of generating the relation tensor $\mathrm{R}$, made up of attention scores that capture relationships between different HD objects $\mathrm{h}_{\mathrm{O}_i}$. These scores are built using a novel hyperdimensional attention mechanism, called *HD-Attention*. Prior work Vaswani et al. (2017); Altabaa et al. (2023) has generated these relational representations using a pairwise inner product between object features in a low dimensional space. In contrast, the HD-Attention mechanism maps object features to a high-dimensional space where (as shown in Menet et al. (2024)), the HD Object hypervectors are quasi-orthogonal, allowing efficient relational representation and object feature superposition in the high dimensional vector space.

The HD-Attention mechanism represents object sequences using the *bundling* operation (i.e., $\oplus$) between HD-encoded sequence elements. Given two objects $\mathrm{O}_i$ and $\mathrm{O}_j$ we first project them onto a hyperspace using the *HD-Encoder* $\phi_{\mathrm{HD}}$. The HD object hypervectors are thus $\mathrm{h}_{\mathrm{O}_i} = \phi_{\mathrm{HD}}(\mathrm{O}_i)$. Before calculating the attention score, these HD objects are made bipolar using the function $\delta(x) = -\mathbb{1}_{\{x<0\}} + \mathbb{1}_{\{x>0\}}$, replacing the binary coordinate-wise majority in the bipolar domain used in (Kanerva, 2022). Thus, the $(i,j)$th element of the relation tensor $\mathrm{R}_{ij}$ denoting the object-level relationships between $\mathrm{O}_i$ and $\mathrm{O}_j$ can be expressed according to the equation 1 where $cos(.)$ denotes the cosine similarity function and $\|\|_2$ denotes the L2 norm:

$$\mathrm{R}_{ij} = \cos(\delta(\mathrm{h}_{\mathrm{O}_i}), \delta(\mathrm{h}_{\mathrm{O}_i} \oplus \mathrm{h}_{\mathrm{O}_j})) = \frac{\langle\delta(\mathrm{h}_{\mathrm{O}_i}), \delta(\mathrm{h}_{\mathrm{O}_i} \oplus \mathrm{h}_{\mathrm{O}_j})\rangle}{D} \tag{1}$$

The denominator of the cosine similarity function $cos(.)$ is $\|\delta(\mathrm{h}_{\mathrm{O}_i})\|_2.\|\delta(\mathrm{h}_{\mathrm{O}_i} \oplus \mathrm{h}_{\mathrm{O}_j})\|_2$. Since the HD objects are bipolar, their L2 norm is $\sqrt{D}$, leading to the expression in Equation 1. We define *bundling* ($\oplus$) as the element-wise real value summation between two bundled HD objects. It captures the *dominant* or *relevant* features of an object pair. The sign of each HD object element follows the sign of the element with a higher magnitude, amplified by dominant features of object pair during training. In step, ❸ the relation tensor matrix $\mathrm{R}$ is normalized using a softmax function to generate $\overline{\mathrm{R}}$. This matrix is used to encode the HD Object hypervectors by mixing them according to their corresponding weights in the normalized relation tensor $\overline{\mathrm{R}}$.

## 6 RESOLVE: ARCHITECTURE OVERVIEW

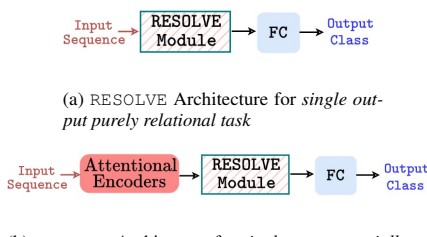

Figure 6: The `RESOLVE` module inside an encoder-decoder for sequence-to-sequence tasks. It includes abstract (turquoise path) and object level (red path) information that are superposed. The `RESOLVE` module operates in a high dimensional space (bold arrows). The rest operate in a low dimensional space (dotted arrows)

The `RESOLVE` module implementation is illustrated in Figure 6, for a sequence-to-sequence encoder-decoder structure with `RESOLVE` Modules (❷ to ❹). An input sequence, in this case a set of tokens, is encoded into embedding vectors and then passed to the Attentional Encoders in Step ❶, which consist of self-attention layers followed by feedforward networks Vaswani et al. (2017). This module is commonly used in sequence-to-sequence modeling and in prior art (Altabaa et al., 2023) to extract object-level information from the sequence.

In step ❷, the output of the Attentional Encoders, which consists of a set of encoded objects, is mapped to a high-dimensional space using the HD Encoder ❷. These HD Object hypervectors are then mixed using the HD-Attention ❸ mechanism to generate $h_{E_O}$. In parallel, a set of relational representations (the learnable symbols of Section 4) S are mapped to high-dimensional space through the same HD-encoder ❷. The resulting hypervectors, denoted by $h_s$, are then combined with the mixed HD Object hypervectors through a *binding* operation in Step ❹. The result is denoted as $h_{S \otimes E_O}$.

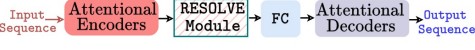

(a) `RESOLVE` Architecture for *single output purely relational task*

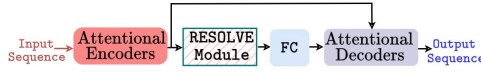

(b) `RESOLVE` Architecture for *single output partially relational task*

(c) `RESOLVE` Architecture for sequence-to- sequence *purely relational task*

(d) `RESOLVE` Architecture for sequence-to- sequence *partially relational task*

Figure 7: `RESOLVE` pipelines for four tasks

High dimensional vectors are known to be holistic Kanerva (2009) meaning that information is uniformly distributed across them. This gives high information redundancy and makes it possible to map to a low-dimensional space with low information loss Yan et al. (2023). The hypervectors gained from Step ❹ ($h_{S \otimes E_O}$) are thus mapped to low-dimensional space through a learnable linear layer in Step ❺. The resulting vectors are then forwarded to a set of Attentional Decoders in Step ❻, which consist of causal-attention and cross-attention layers Vaswani et al. (2017).

Figure 7 shows four different `RESOLVE` architecture configurations used for different tasks. The `RESOLVE` architectures illustrated in (Figure 7a) consists of a single `RESOLVE` encoder followed by a fully connected layer. It is used for single output purely relational tasks (e.g. pairwise ordering) that don't require an attentional object level encoding. On the other hand, Figure 7b shows the same architecture with an attentional encoder in the front-end used to process object level features for single output partially relational tasks (e.g. learning the abstract rule of subtraction). Figure 7c and 7d shows `RESOLVE` architecture for sequence-to-sequence purely (e.g. sorting) and partially relational tasks (e.g. mathematical problem solving (Saxton et al., 2019)) respectively. Both of them use an attentional encoder to process object level features and an attentional decoder to generate the output sequence. However, the architecture in Figure 7d requires a skip-connection between the encoder and the decoder because the output sequence in the partially relational tasks relies on object features as well as relational representations.

# 7 EXPERIMENTS

We have evaluated the performance of RESOLVE compared to the state-of-the-art on several relational tasks: (1) Single output purely relational tasks (pairwise ordering, a sequence of image pattern learning with preprocessed inputs); (2) Single output partially relational tasks (sequence of image pattern learning with low-processed inputs, mathematical abstract rule learning from images); (3) Sequence-to-sequence purely relational tasks (Sorting); (4) Sequence-to-sequence partially relational tasks (Mathematical problem solving). The baselines used for comparison are CorelNet Kerg et al. (2022) with Softmax activation, Predinet Shanahan et al. (2020), the Abstractor Altabaa et al. (2023), the transformer Vaswani et al. (2017), a multi-layer-perceptron (as evaluated in Altabaa et al. (2023)) and the LEN Zheng et al. (2019), a neuro symbolic architecture.

## 7.1 SINGLE OUTPUT PURELY RELATIONAL TASKS

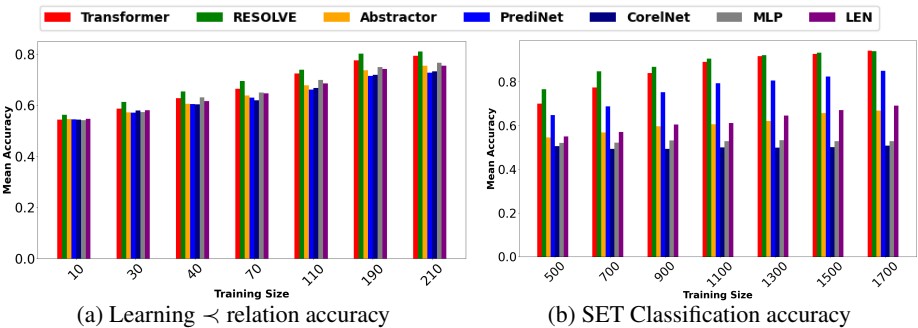

(a) Learning $\prec$ relation accuracy     (b) SET Classification accuracy

Figure 8: Experiments on single output *purely* relational tasks and comparison to SOTA.

**Order relations: modeling asymmetric relations** As described in Altabaa et al. (2023), we generated 64 random objects represented by iid Gaussian vectors $o_i \sim \mathcal{N}(0, I) \in \mathbb{R}^{32}$, and established an anti-symmetric order relation between them $o_1 \prec o_2 \prec \cdots \prec o_{64}$. From 4096 possible object pairs $(o_i, o_j)$, 15% are used as a validation set and 35% as a test set. We train models on varying proportions of the remaining 50% and evaluate accuracy on the test set, conducting 5 trials for each training set size. The models must generalize based on the transitivity of the $\prec$ relation using a limited number of training examples. The training sample sizes range between 10 and 210 samples. Figure 8a demonstrates the high capability of RESOLVE to generalize with just a few examples, achieving over 80% accuracy with just 210 samples ($1.05\times$ better than the second best model and $1.09\times$ better than Abstractor). The Transformer model is the second best performer, better than the Abstractor and CorelNet-Softmax due to the lower level of abstraction needed for learning asymmetric relations.

*SET*: modeling multi-dimensional relations with *pre*-processed objects
In the SET (Altabaa et al., 2023) task, players are presented with a sequence of cards. Each card varies along four dimensions: color, number, pattern and shape. A triplet of cards forms a "set" if they either all share the same value or each have a unique value (as in Figure 9). The task is to classify triplets of card images as either a "set" or not. The shared architecture for processing the card images in all baselines as well as RESOLVE is CNN $\rightarrow \{\cdot\} \rightarrow$ Flatten $\rightarrow$ Dense, where $\{\cdot\}$ is one of the aforementioned modules. The CNN embedder is pre-trained and object features are taken from the last linear layer of the model. The relational models thus focus on learning the abstract rules without having to encode object features. For this specific task, there are four relational representations (e.g., shape, color, etc.) and one abstract rule (whether it is a triplet or not).

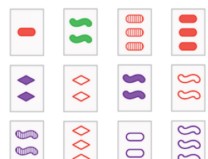

Figure 9: The SET game

Figure 8b shows RESOLVE outperforms all the baselines (up to $1.05x$ better than the second best model and $1.11x$ better than the Abstractor), as it balances object features with relational representations. In this particular case, PrediNet also shows high accuracy. Its feature vectors are less connected to object-level features than those of the transformer but more than those of the Abstractor. This experiment shows that for descriminative purely relational tasks, abstract rules are often easy to extract and are highly correlated to object features, resulting in the transformer outperforming the Abstractor.

## 7.2 SINGLE OUTPUT PARTIALLY RELATIONAL TASKS

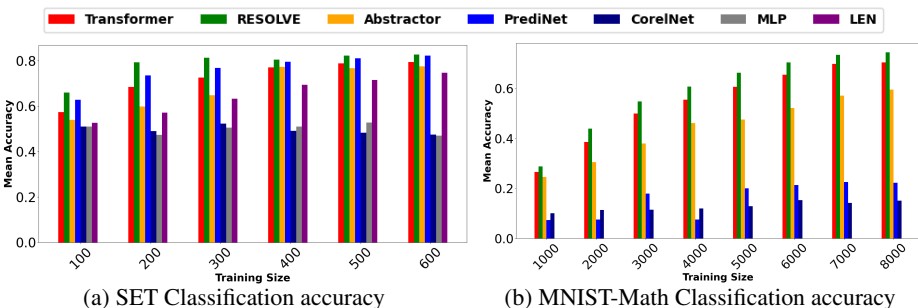

(a) SET Classification accuracy      (b) MNIST-Math Classification accuracy

Figure 10: Experiments on single output *partially* relational tasks and comparison to SOTA.

***SET*: modeling multi-dimensional relations with *low*-processed objects** Instead of extracting highly encoded object level features from the pre-trained CNN used in Section 7.1, we extract the feature map of the first convolutional layer of the pretrained CNN to assess the ability of RESOLVE to handle low processed object level features. Figure 10a shows the mean accuracy of different relational models when trained on small portion of the dataset. RESOLVE outperforms the state of the art with more than $80\%$ accuracy using just 600 training samples. In contrast to the Section 7.1, PrediNet is the second best model thanks to its balanced trade-off between object-level feature processing and abstract feature encoding.

***MNIST-MATH*: extracting mathematical rules from a pair of digit images** In this case (Figure 11), the math rule to extract is a non-linear weighted subtraction (i.e, $F(a, b) = |3a - 2b|$). This task is partially relational since the input image label is unknown, making object level feature extraction more critical. According to Figure 10b, RESOLVE outperforms the other baselines, with $1.14\times$ better accuracy than the transformer and $1.47\times$ better accuracy than the Abstractor. The transformer outperforms the Abstractor here due to to the relative simplicity of the abstract rule, this problem relies more on object level information than abstract information.

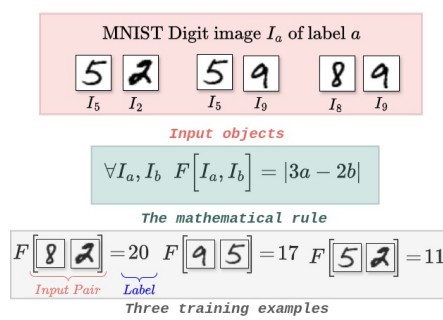

Figure 11: MNIST-Math classification task

## 7.3 OBJECT-SORTING: PURELY RELATIONAL SEQUENCE-TO-SEQUENCE TASKS

We generate have generated random objects for the sorting task. First, we create two sets of random attributes: $\mathcal{A} = a_1, a_2, a_3, a_4$, where and $\mathcal{B} = b_1, \ldots, b_{12}$. Each set of attributes has a strict ordering: $a_1 \prec a_2 \prec a_3 \prec a_4$ for $\mathcal{A}$ and $b_1 \prec b_2 \prec \cdots \prec b_{12}$ for $\mathcal{B}$. Our random objects are formed by taking the Cartesian product of these two sets, $\mathcal{O} = \mathcal{A} \times \mathcal{B}$, resulting in $N = 48$ objects. Each object in $\mathcal{O}$ is a vector in $\mathbb{R}^{12}$, formed by concatenating one attribute from $\mathcal{A}$ with one attribute from $\mathcal{B}$.

We then establish a strict ordering relation for $\mathcal{O}$, using the order relation of $\mathcal{A}$ as the primary key and the order relation of $\mathcal{B}$ as the secondary key. Specifically, $(a_i, b_j) \prec (a_k, b_l)$ if $a_i \prec a_k$ or if $a_i = a_k$

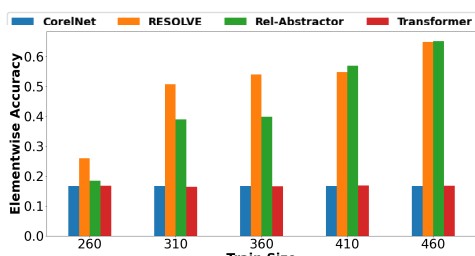

Figure 12: Performance of RESOLVE compared to baselines for 6 elements sequence sorting.

and $b_j \prec b_l$. We generated a randomly permuted set of 5 and a set of 6 objects in $\mathcal{O}$. The target sequences are the indices representing the sorted order of the object sequences (similar to the 'argsort' function). The training data is uniformly sampled from the set of 6 elements based sequences in $\mathcal{O}$. We generate non-overlapping validation and testing datasets in the following proportion: $20\%$ testing, $10\%$ validation and $70\%$ training.

We used *element wise accuracy* to assess the performance of RESOLVE, as in (Altabaa et al., 2023). The accuracy of RESOLVE is compared against the Relational Abstractor (Altabaa et al., 2023), Transformer Vaswani et al. (2017) and CorelNet(Kerg et al., 2022).

Figure 12 shows that RESOLVE achieves better accuracy than the baselines (1.56x to 1.02x better than Relational-Abstractor). Relational-Abstractor (Altabaa et al., 2023) still outperforms the transformer and CorelNet-Softmax, validating the results of (Altabaa et al., 2023). RESOLVE demonstrates a high generalizability compared to SOTA. However, as the number of training sample increases, Relational Abstractor and RESOLVE converge toward the same level of accuracy with increased training data.

## 7.4 MATH PROBLEM-SOLVING: PARTIALLY-RELATIONAL SEQUENCE-TO-SEQUENCE TASKS

Task: Numbers_place_value
Question: what is the tens digit of 3585792?
Answer: 9

Task: Comparison_pair
Question: Which is bigger:  4/37 or 7/65?
Answer: 4/37 is bigger

Figure 13: Examples of input/target sequences from the math problem-solving dataset.

| | | Comparison Closest | | | | Comparison Pair | | | | Comparison Place Value | | | | |
|---|---|---|---|---|---|---|---|---|---|---|---|---|---|---|
| | *Train size* | *100* | *1000* | *10000* | *avg* | *100* | *1000* | *10000* | *avg* | *100* | *1000* | *10000* | *avg* | *Overall* |
| | RESOLVE | **15.08** | **20.88** | 52.36 | ***29.44*** | 28.43 | **38.15** | 66.84 | ***44.47*** | 19.36 | **36.98** | 98.68 | *51.67* | **41.86** |
| Model | Rel-Abstractor | 13.77 | 19.49 | **52.46** | 28.57 | 26.86 | 35.82 | **69.19** | 43.95 | 19.93 | 32.21 | 99.43 | 50.52 | 41.01 |
| | Transformer | 14.25 | 18.08 | 37.76 | 23.36 | 30.73 | 34.1 | 64.37 | 43.06 | 21.1 | 32.91 | **99.64** | 51.21 | 39.21 |

Table 1: Accuracy (probability of correct answer) of RESOLVE compared to SOTA for three mathematical reasoning tasks (*best accuracy in bold*)

We further evaluate RESOLVE on a mathematical reasoning dataset (Figure 13), which represents a *partially relational* sequence-to-sequence problem. Table 1 presents the accuracy achieved by the relational abstractor, transformer, and RESOLVE on three different datasets using 100 to 10,000 training samples. The accuracy corresponds to the percentage of full sequence matches, each one representing a correct answer. We report the average accuracy across the three different training sizes in the table, as well as an overall accuracy for all test cases. RESOLVE outperforms the state-of-the-art (SOTA) on average across the three test cases. It also achieves higher accuracy with a small training set, demonstrating the generalizability of the proposed architecture. Notably, neither the relational representation nor the object-level features alone are sufficient for inducing abstract rules from partially relational tasks, which penalizes both the Abstractor and transformer. In contrast, RESOLVE combines both levels of knowledge into a single structure.

## 7.5 COMPUTATIONAL OVERHEAD ASSESSMENT

| Embedding size | | *32* | | | *64* | |
|---|---|---|---|---|---|---|
| Model | $\pi$ | $\beta$ (L1 Cache) | $\beta$ (DRAM) | $\pi$ | $\beta$ (L1 Cache) | $\beta$ (DRAM) |
| HD Attention | **1.99** | **0.787** | 0.867 | **1.99** | **0.788** | **0.870** |
| Self-Attention | 1.99 | 0.783 | **0.869** | 1.97 | 0.781 | 0.863 |

Table 2: Comparison between the HD-Attention Self attention mechanism in term of computational overhead. $\beta$ is the bandwidth bound in *Flop/Byte* and $\pi$ is processor peak performance in *GFLOPS*

We assessed the computational overhead of the HD-Attention mechanism described in Section 5 against the baseline of a regular self attention mechanism Vaswani et al. (2017). The operations are done on a CPU using Multi-threading. The memory overhead is measured at the level of DRAM and L1 cache memory using the roofline model Ofenbeck et al. (2014). A high $\beta$ value means that the system is less likely to encounter memory bottlenecks. A high $\pi$ means the processor is capable of performing more computations per cycle. Table 2 shows that the HD-Attention mechansim has better computational performance compared to self-attention ($\pi$) with higher memory bandwidth $\beta$. This is due to the use of the bipolar HD representation and operations such as bundling and binding.

## 8 CONCLUSION

In this work we have presented RESOLVE, a vector-symbolic framework for relational learning that outperforms the state of the art thanks to its use of high-dimensional attention mappings for mixing relational representations and object features. In future we plan to examine multimodal learning tasks and sequence-to-sequence learning tasks in the high-dimensional domain, taking advantage of the computational efficiency of vector-symbolic architectures.

## ACKNOWLEDGEMENTS

Acknowledgements removed for review.

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

## A  APPENDIX

### CODE AND REPRODUCIBILITY

The code, detailed experimental logs, and instructions for reproducing our experimental results are available at: `https://github.com/mmejri3/RESOLVE`.

### SINGLE OUTPUT TASKS

In this section, we provide comprehensive information on the architectures, hyperparameters, and implementation details of our experiments. All models and experiments were developed using TensorFlow. The code, along with detailed experimental logs and instructions for reproduction, is available in the project's public repository.

### A.1 COMPUTATIONAL RESOURCES

For training the RESOLVE and SOTA models on the single-output relational tasks, we used a GPU (Nvidia RTX A6000 with 48GB of RAM). For training LARS-VSA and SOTA on purely and partially sequence-to-sequence abstract reasoning tasks, we used a single GPU (Nvidia A100 with 80GB of RAM). The overhead assessment of the HD-attention mechanism was conducted on a CPU (11th Gen Intel® Core™ i7).

### A.2 *Single Output Purely Relational Tasks*

#### A.2.1 PAIRWISE ORDER

Each model in this experiment follows the structure: `input → module → flatten → MLP`, where `module` represents one of the described modules, and `MLP` is a multilayer perceptron with one hidden layer of 32 neurons activated by ReLU.

**RESOLVE Architecture** Each model consists of a single module and a hypervector of dimensionality $D = 1024$. We use a dropout rate of 0.1 to prevent overfitting. The two hypervectors are flattened and passed through hidden layers containing 32 neurons with ReLU activation, followed by a final layer with one neuron activated by a sigmoid function.

**Abstractor Architecture** The Abstractor module utilizes the following hyperparameters: number of layers $L = 1$, relation dimension $d_r = 4$, symbol dimension $d_s = 64$, projection (key) dimension $d_k = 16$, feedforward hidden dimension $d_{\text{ff}} = 64$, and relation activation function $\sigma_{\text{rel}} = \text{softmax}$. No layer normalization or residual connections are applied. Positional symbols, which are learned parameters, are used as the symbol assignment mechanism. The output of the Abstractor module is flattened and passed to the `MLP`.

**CoRelNet Architecture** CoRelNet has no hyperparameters. Given a sequence of objects, $X = (x_1, \ldots, x_m)$, standard CoRelNet (Kerg et al., 2022) computes the inner product and applies the Softmax function. We also add a learnable linear map, $W \in \mathbb{R}^{d \times d}$. Hence, $\bar{R} = \text{Softmax}(R)$, where $R = [\langle Wx_i, Wx_j \rangle] \, ij$. The CoRelNet architecture flattens $\bar{R}$ and passes it to an `MLP` to produce the output. The asymmetric variant of CoRelNet is given by $\bar{R} = \text{Softmax}(R)$, where $R = [\langle W_1 x_i, W_2 x_j \rangle] \, ij$, and $W_1, W_2 \in \mathbb{R}^{d \times d}$ are learnable matrices.

**PrediNet Architecture** Our implementation of PrediNet (Shanahan et al., 2020) is based on the authors' publicly available code. We used the following hyperparameters: 4 heads, 16 relations, and a key dimension of 4 (see the original paper for the definitions of these hyperparameters). The output of the PrediNet module is flattened and passed to the `MLP`.

**MLP** The embeddings of the objects are concatenated and passed directly to an `MLP`, which has two hidden layers, each containing 32 neurons with ReLU activation.

**Training/Evaluation** We use cross-entropy loss and the AdamW optimizer with a learning rate of $10^{-4}$. The batch size is 128, and training is conducted for 100 epochs. Evaluation is performed on the test set. The experiments are repeated 5 times, and we report the mean accuracy and standard deviation.

#### A.2.2 *SET*

The card images are RGB images with dimensions of $70 \times 50 \times 3$. A CNN embedder processes these images individually, producing embeddings of dimension $d = 64$ for each card. The CNN is trained to predict four attributes of each card. After training, embeddings are extracted from an intermediate layer, and the CNN parameters are frozen. The common architecture follows the structure: `CNN Embedder → Abstractor, CoRelNet, PrediNet, MLP → Flatten → Dense(2)`. Initial tests with the standard CoRelNet showed no learning. However, removing the Softmax activation improved performance slightly. Hyperparameter details are provided below.

**Common Embedder Architecture:** The architecture follows this structure: `Conv2D →` `MaxPool2D → Conv2D → MaxPool2D → Flatten → Dense(64, ReLU) → Dense(64,` `ReLU) → Dense(2)`. The embedding is taken from the penultimate layer. The CNN is trained to perfectly predict the four attributes of each card, achieving near-zero loss.

**RESOLVE Architecture:** The RESOLVE module has the following hyperparameters: hypervector dimension $D = 1024$. The outputs are flattened and passed through a feedforward hidden layer with dimension $d_{\text{ff}} = 64$, followed by a final layer with a single neuron and sigmoid activation. A dropout rate of 0.4 is used to prevent overfitting.

**Abstractor Architecture:** The Abstractor module uses the following hyperparameters: number of layers $L = 1$, relation dimension $d_r = 4$, symmetric relations ($W_q^i = W_k^i$ for $i \in [d_r]$), ReLU activation for relations, symbol dimension $d_s = 64$, projection (key) dimension $d_k = 16$, feedforward hidden dimension $d_{\text{ff}} = 64$, and no layer normalization or residual connections. Positional symbols, which are learned parameters, are used as the symbol assignment mechanism.

**CoRelNet Architecture:** In this variant of CoRelNet, we found that removing the Softmax activation improved performance. The standard CoRelNet computes $R = \text{Softmax}(A)$, where $A = [\langle W x_i, W x_j \rangle]_{ij}$.

**PrediNet Architecture:** The hyperparameters used are 4 heads, 16 relations, and a key dimension of 4, as described in the original paper. The output of the PrediNet module is flattened and passed to the MLP.

**MLP:** The embeddings of the objects are concatenated and passed directly to an MLP with two hidden layers, each containing 32 neurons with ReLU activation.

**Data Generation:** The dataset is generated by randomly sampling a "set" with probability 1/2 and a non-"set" with probability 1/2. The triplet of cards is then randomly shuffled.

**Training/Evaluation:** We use cross-entropy loss and the AdamW optimizer with a learning rate of $10^{-4}$. The batch size is 512, and training is conducted for 200 epochs. Evaluation is performed on the test set. We train our model on a randomly sampled set of $N$ samples, where $N \in 500, 700, 900, 1100, 1300, 1500, 1700$.

### A.3 *Single Output Partially Relational Tasks*

### A.3.1 *SET*

We used the same settings as in the previous SET experiment. However, in this task, the input features used as a sequence of objects are derived from the first convolutional layer of the pre-trained `CNN`. This approach avoids using highly processed object-level features, allowing us to assess the ability of RESOLVE and the baseline models to capture both object-level features and relational representations.

We did not change the hyperparameters of the baseline models or the RESOLVE model. However, we added an attentional encoder at the front end, with a single layer and two heads.

### A.3.2 *MNIST-MATH*

This experiment is inspired by the MNIST digits addition task introduced by Manhaeve et al. (2018). We randomly selected 10,000 pairs of MNIST digits from the MNIST training set and generated labels using a non-linear mathematical formula: $F(a, b) = |3a - 2b|$.

The digits are normalized and flattened before being passed to the relational models. We used the same hyperparameters as in the SET experiment.

### A.4 RELATIONAL SEQUENCE-TO-SEQUENCE TASKS

### A.4.1 *Object-Sorting Task*

**RESOLVE Architecture** We used architecture (c) from Figure 7. The encoder includes a Batch-Normalization layer. The RESOLVE architecture consists of a single module with a hyperdimensional

dimension of $D = 1024$. The decoder has 4 layers, 2 attention heads, a feedforward network with 64 hidden units, and a model dimension of 64.

**Abstractor Architecture**    Each of the Encoder, Abstractor, and Decoder modules consists of $L = 2$ layers, with 2 attention heads/relation dimensions, a feedforward network with $d_{\text{ff}} = 64$ hidden units, and a model/symbol dimension of $d_{\text{model}} = 64$. The relation activation function is $\sigma_{\text{rel}} = \text{Softmax}$. Positional symbols are used as the symbol assignment mechanism, which are learned parameters of the model.

**Transformer Architecture**    We implemented the standard Transformer architecture as described by (Vaswani et al., 2017). Both the Encoder and Decoder modules share the same hyperparameters, with an increased number of layers. Specifically, we use 4 layers, 2 attention heads, a feedforward network with 64 hidden units, and a model dimension of 64.

**Training and Evaluation**    The models are trained using cross-entropy loss and the Adam optimizer with a learning rate of $5 \cdot 10^{-4}$. We use a batch size of 128 and train for 500 epochs. To evaluate the learning curves, we vary the training set size, sampling random subsets ranging from 260 to 460 samples in increments of 50. Each sample consists of an input-output sequence pair. For each model and training set size, we perform 10 runs with different random seeds and report the mean accuracy.

A.4.2 *Math Problem-Solving*

The dataset consists of various math problem-solving tasks, each featuring a collection of question-answer pairs. These tasks cover areas such as solving equations, expanding polynomial products, differentiating functions, predicting sequence terms, and more. The dataset includes 2 million training examples and 10,000 validation examples per task. Questions are limited to a maximum length of 160 characters, while answers are restricted to 30 characters. Character-level encoding is used, with a shared alphabet of 95 characters, which includes upper and lower case letters, digits, punctuation, and special tokens for start, end, and padding.

**Abstractor Architectures**    The Encoder, Abstractor, and Decoder modules share identical hyperparameters: number of layers $L = 1$, relation dimension/number of heads $d_r = n_h = 2$, symbol dimension/model dimension $d_s = d_{\text{model}} = 64$, projection (key) dimension $d_k = 32$, and feedforward hidden dimension $d_{\text{ff}} = 128$. The relation activation function in the Abstractor is $\sigma_{\text{rel}} = \text{Softmax}$. One model uses positional symbols with sinusoidal embeddings, while the other uses symbolic attention with a symbol library of $n_s = 128$ learned symbols and 2-head symbolic attention.

**Transformer Architecture**    The Transformer Encoder and Decoder have the same hyperparameters as the Encoder and Decoder in the Abstractor architecture.

**RESOLVE Architectures**    The RESOLVE model follows architecture (D) from Figure 7. We use the same Decoder as the Abstractor architecture. The RESOLVE model has a single module and a hyperdimensional dimension of $D = 1024$.

**Training and Evaluation**    Each model is trained for 1000 epochs using categorical cross-entropy loss and the Adam optimizer with a learning rate of $6 \times 10^{-4}$, $\beta_1 = 0.9$, $\beta_2 = 0.995$, and $\varepsilon = 10^{-9}$. The batch size is 64. The training set consists of $N$ samples, where $N \in 100, 1000, 10,000$.

