# OpenReview forum: "RESOLVE: Relational Reasoning with Symbolic and Object-Level Features Using Vector Symbolic Processing"
_ICLR.cc/2025/Conference — ICLR 2025 Conference Withdrawn Submission_

### Official Review · Reviewer_5Cje · 2024-10-28

**Soundness:** 2
**Presentation:** 3
**Contribution:** 2
**Rating:** 3
**Confidence:** 4

**Summary:**

This work proposes a novel attention module, HD-Attention with a RESOLVE neuro-vector symbolic architecture to solve the relational reasoning problems. Experiments have been conducted on four tasks: relational classification, partial relational classification, sorting, and math problems. Results show that RESOLVE not only achieves low compute latency and memory efficiency, but offers better accuracy on these tasks than previous baselines.

**Strengths:**

This work proposes RESOLVE architecture, which is a includes reasonable modules and strategies for relational reasoning problems. Meanwhile, the tasks of experiments are diverse to illustrate the effectiveness of RESOLVE. The writing is good and very easy to understand.

**Weaknesses:**

The contributions of this paper do not seem solid.

(1) Based on the Figure 4, it is not clear that why the RESOLVE's architecture is  better than Transformer and Abstractor's. It can be seen that HD-Attention is more complex, but its better performance in relational reasoning does not seem intuitive.

(2) From experiments, RESOLVE does not seem to show significant improvements in performance and efficiency compared with Abstractor.

(3) There is no ablation study or interpretability analysis to demonstrate the effectiveness of HD-Attention.

**Questions:**

(1) Can you further illustrate the advantages of RESOLVE compared with Transformer and Abstractor? Or can you provide some explainable cases to prove the relational reasoning ability of RESOLVE?

(2) Why doesn't RESOLVE show significant improvement compared with Abstractor?

---

### Official Review · Reviewer_Nvu8 · 2024-11-01

**Soundness:** 1
**Presentation:** 3
**Contribution:** 1
**Rating:** 3
**Confidence:** 4

**Summary:**

This paper presents an architecture called *RESOLVE*, aiming to integrate relational representations with object-level features through a "hyper-dimensional" vector representation. A core idea of the architecture is to map the input vectors into a high-dimensional space (1-2 orders of magnitude larger), and perform computation in this high-dimensional space. The resulting module is called the "Hyper Dimensional Encoder", and is a variant of a Transformer encoder, where you: 1) map to a higher("hyper")-dimensional space, 2) compute attention on these high-dimensional vectors, and 3) compute a Hadamard product between the outputs of the attention and a set of learnable parameters. The attention operation is modified to first apply a sign function to the "hyper-dimensional" vectors so that they are in $\\{-1, +1\\}^{D}$. The authors interpret the attention component of this operation as computing "object-level features" and the Hadamard product with the learnable parameters as representing "relational" information.

This work draws heavily from a recently proposed architecture called the *Abstractor*.

**Strengths:**

- This paper aims to incorporate ideas from hyper-dimensional computing into recently-proposed ideas on relational architectures, which is a novel and promising direction.
- The organization and presentation of the paper are generally clear, with many figures used to explain the proposed ideas. I particularly liked the numbered annotations in the figures that are then referred to in the text. It makes it very easy to understand which part of the figure is being discussed.

**Weaknesses:**

On the conceptual aspects of the paper:
- *It is not clear how this architecture captures relational information.* The proposed architecture involves (roughly): 1) computing self-attention on vector embeddings that are projected to a higher dimension, and 2) multiplying the result with a set of learnable vectors, called "symbols". According to the authors, the attention operation (1) captures object-level features, while the Hadamard product with learnable symbol vectors (2) is intended to capture relational information. However, the symbols are *input-independent*; they don't capture any features of the input, including any notion of "relational information". There are no comparisons being computed in the symbols. They act more as positional embeddings. This seems to stem from a misunderstanding of the Abstractor. In the Abstractor, the *relation tensor* captures relational information, not the symbols. The symbols merely act as pointers to refer to objects in the context and the relational representation is computed by binding the relations in the relation tensor to the symbols via a convex combination. The symbols don't inherently represent relational information (or any information for that matter, besides positional information to act as pointers or identifiers).
- *The motivation of the architecture as an improvement on partially-relational settings is unclear*. The paper says that the Abstractor faces challenges in so-called "partially-relational tasks" and suggests that *RESOLVE* addresses these issues. Can the authors elaborate on this? The Abstractor paper also tackles partially relational tasks, using an architecture that integrates a standard encoder with the Abstractor, which is also used by *RESOLVE* in the partially relational experiments.
- *HD-Attention appears to be non-differentiable*. In the proposal of "Hyper-Dimensional Attention", a $\mathrm{sign}(\cdot)$ function is applied before computing a cosine similarity. The sign function has zero gradient almost everywhere. Doesn't this make the overall architecture non-differentiable? E.g., gradients can't propagate to $\phi_{MD}$ or previous layers?

On the experimental evaluation:
- Do the experiments control for parameter count in the comparisons? What are the parameter counts of different models in the comparison? Since the *RESOLVE* architecture involves projecting up to a high-dimensional space that is 1-2 orders of magnitude larger than the latent space of the baseline models, it is important to control for model size and computational cost when performing a comparison.
- Some of the reported experimental results do not agree with the results reported in the Abstractor paper. For example, for *SET*, eye-balling your figure and there's: in their figure, the Abstractor achieves ~90% acc at 1000 training examples and nearly 100% at 2000, whereas in your figure the Abstractor is below 60%. What explains this discrepancy? How are the hyperparameters of the different models chosen? Is a hyperparameter search performed for each model?
- In the object-sorting experiments, the sequence length is only 6, which is smaller than what is considered in the Abstractor paper. Why was the sequence length decreased in your experiments? How does *RESOLVE* compare to the baselines at longer sequence lengths?
- On the mathematical problem-solving experiments, only three tasks are evaluated, and a different set of tasks to the ones considered in the Abstractor paper are chosen. How does *RESOLVE* perform on other tasks in the dataset? Why did you choose to change the set of tasks?
- The Abstractor paper evaluates two types of symbol assignment mechanisms: one is positional symbols (which is what *RESOLVE* seems to use) and the other is "symbolic attention". On the mathematical problem-solving experiments in the Abstractor paper, the latter has significantly stronger performance. Which version is used in your experiments?
- One of the claims about the proposed architecture is computational efficiency, and in section 7.5 the authors assess "computational overhead" in terms of L1 cache and DRAM usage on a CPU. It was unclear to me what exactly is being claimed here and what is being evaluated. The numbers look nearly identical. How should this be interpreted?
- How does *RESOLVE* compare to the baselines in terms of training speed on a GPU? Would the increased dimensionality imply *slower* training speed?

Other feedback:
- It would be useful to incorporate a brief background section on "vector-symbolic architectures" and "hyperdimensional computing", to explain the aspects of these ideas that are relevant to the paper.
- Some figures are low-resolution. It would be nice to include high-resolution renderings of these figures in a future version of the paper.

**Questions:**

See above.

Also:
- In what sense does the Hadamard product with the learnable symbol vectors represent "relational information"?
- What role does the "hyper-dimensional" projection $\phi_{MD}$ play versus the hadamard product with symbols? E.g., if you maintain HD-attention but remove the Hadamard product, how does that affect performance? What about the other way around?
- In lines 212-215, you say that the query/key/values in HD-Attention are identical. Does this mean there are no $W_Q, W_K, W_V$ projections? Can you explain this choice? In standard attention and Transformers, those parameters are crucial to enabling powerful multi-layer computational circuits.
- In lines 149-152, you say that models based on the relational bottleneck principle suffer from interference between relational representations and object features in deeper layers. Can you clarify what you mean by this?
- MD-attention uses the cosine similarity as the comparison function, which is bounded in range in [-1,1]. This means that the attention scores cannot be sharp. In particular, as the number of objects in the context increases, the entropy necessarily increases and the distribution tends towards uniform. Could you comment on this issue?

---

### Official Review · Reviewer_kimi · 2024-11-02

**Soundness:** 3
**Presentation:** 2
**Contribution:** 3
**Rating:** 6
**Confidence:** 2

**Summary:**

This paper introduces RESOLVE, a neuro-symbolic architecture combining object-level features with relational representations in high-dimensional spaces to perform both object and relational-level reasoning.  By exploiting efficient operations RESOLVE allows both object and relational representations to coexist without interferring. Moreover, a novel attention mechanism is introduced that shows a fast computation wrt sota techniques, as well as showing better generalization with higher accuracy in pure relational reasoning tasks.

**Strengths:**

- RESOLVE faces a very interesting task.
- The methodology builds upon existing ideas, however the framework is novel and present several advantages.
- RESOLVE is shown to perform better than existing approaches on a wide variety of tasks.

**Weaknesses:**

- Related work on NeSy should be significantly improved. In the same line, very different approaches are listed, but these approaches rely on very different principles. For instance, DeepProbLog and LTN require the rules to be already given. DeepProbLog can learn (ONLY) the rule confidence (aka parameter learning), while LTN is not learning nor the rules nor their weights as far as I know. RCBMs (Barbiero et al.) are learning the rules when using DCR [1] as task predictor, which is very different from the others. However they require a template (like an inductive bias) to learn the rules. Hence it is totally confusing mixing these works together. Similarly for putting the semantic loss in the hotchpotch.
Moreover, concerning rule learning, a wide class of methodologies is not discussed. For instance, there have been many proposed systems for relational setting with KGE, e.g. AMIE [2], RNNLogic [3], NCRL [4], only to mention some. In addition the whole area of inductive logic programming, which has recently seen a novel advancement.
In summary I think the related work section on rule learning should be significantly improved and better clarified how the proposed work differ/contextualize wrt existing ones.

- Even if there are many figures, examples, detailed comparisons with existing works (sect 3,4,5), I found the presentation quite confusing and intricate. Personally, I would re-elaborate a bit the flow of the paper to make it more clear the different aspects of the method.

OTHER COMMENTS:
- I think it would be useful if the authors clarify the examples in the figures 1 and 2. For instance, what does it mean ">" in Fig. 1? Also intuitively, should not be that "m" is dominant over "n" if "m>n" instead of the opposite? Also in Fig. 2, I didn't get exactly the role of object-features wrt relational information. I understood that the points of these figures are to explain the difference between purely and partially relational tasks, hence it would have been more useful to keep the same example, but using the objects/relations in different ways for different (pure vs partial) tasks.
- Section 3 is very interesting, but I think the name "Overview" is a bit confusing or too general. What you're actually doing is comparing in more detail your methodology (even if it has not been formally defined yet) with closest related works such as transformers and the abstractor. Hence you're still somehow discussing the related work, or if you prefer the background of your method. Hence, I would add section 3 as a subsection of section 2, making it clear from the beginning this zoom, and that you're focusing on methodologies for transformer-based architectures.
- Section 4 does something similar but wrt the attention types. Hence again, I would put it as a separate subsection of Related work. Alternatively, I'd move both current sections 3 and 4 after the definition of your model's architecture in Section 6 (so that readers are already familiar with what you propose, as a more detailed analysis of the differences wrt existing approaches. Moreover, note that in Section 4 is not very clear when the caption ends and starts the main text. Please keep them more separate.
- "We generate have generated" typo


References:
[1] Barbiero, Pietro, et al. "Interpretable neural-symbolic concept reasoning." International Conference on Machine Learning. PMLR, 2023.
[2] Galárraga, Luis Antonio, et al. "AMIE: association rule mining under incomplete evidence in ontological knowledge bases." Proceedings of the 22nd international conference on World Wide Web. 2013.
[3] Qu, Meng, et al. "RNNLogic: Learning Logic Rules for Reasoning on Knowledge Graphs." International Conference on Learning Representations.
[4] Cheng, Keiwei, Nesreen K. Amed, and Yizhou Sun. "Neural Compositional Rule Learning for Knowledge Graph Reasoning." International Conference on Learning Representations (ICLR). 2023.

**Questions:**

1) "relational bottleneck problem (capturing relational information between data/objects rather than input data/object attributes or features from limited training data)", this seems like the difference between pure symbolic data vs sub-symbolic data (like standard data representation for KGE vs GNNs). Is this what you mean by relational bottleneck problem?
2) Can the authors show some examples of learnt abstract rules in the different tasks?

---

### Official Review · Reviewer_Dz6W · 2024-11-04

**Soundness:** 2
**Presentation:** 1
**Contribution:** 2
**Rating:** 3
**Confidence:** 4

**Summary:**

This paper presents RESOLVE, a method to train transformers-based architectures to solve relational reasoning tasks. The authors argue that the recently introduced *Abstractor* module allows only for "*pure relational reasoning*", and lack "*partial relational reasoning*". Thus, abstractor-equipped model could solve reasoning tasks in which the object on which reasoning is perform is lacking concrete semantic meaning (*e.g.* learning how to order image-based objects, independent of any semantic meaning in these objects), but would lack the ability to perform math based on MNIST inputs (in which the images' content contains the semantic of the digits).
To do so, they use both deep and symbolic encodings., combined through their HD attention mechanism, that learn vector-based representation of relations.

**Strengths:**

**The problem is relevant.** Symbolic reasoning is lacking in transformer based architectures.

**Many forms of explanations are provided.** The authors outline several times, with schematic drawings, the intuition behind their methods.

There is overall a lot of work to make this paper a good contribution to the ICLR community, but I believe that the provided feedback can help the authors improve their presentation.

**Weaknesses:**

**Confusing structure**. The paper should be structured in the following way to make it easy for the reader to understand the methods and exact contributions:
* Introduction (the problem at hand and the intuition about proposed solution, which is well done here).
* Background (on which your methods build upon: Transformers, Abstractors, ...).
* Your method (details about RESOLVE)
* Experimental evaluations (with first the details on the evaluation method, e.g. what dataset did you use, for how many epochs, do you report training, testing or validation accuracies, how the data was splitted, why these datasets, ... etc.). Then precise scientific questions:
Q1: Can RESOLVE outperform the existing baselines on pure relational reasoning tasks ?
Q2: On partial RR tasks ?
Q3: Can it use learn faster ?
Q4: What are the core components (ablation study e.g. without HD)
* Related Work (other approaches to solve related problems)
* Conclusion and future work.

**Poor figures**. The figures are not clear and neat. Please provide vectorial (svg or pdf format) figures. Each figure outlines one point. For example, the first two figures can be merged, you want to highlight the contrast between the two reasoning tasks. Explicitly show the difference on the figures between the two.

The figures and tables can thus help the readers during their first pass over the paper. They should thus be structuring the paper like, e.g.:
1/ Figure 1 (describing the problem, if the problem is not obvious),
2/ Figure 2 (describing the method), highlighting its core contribution.
3/ Figures and Tables describing each important results (that answers precise scientific questions mentioned above).

Further, each caption of each figure should be built in the following way:
The first sentence should highlight the main message of the Figure/Table (e.g. "Our method outperforms the existing SOTA methods on the studied problem.")
The next sentences then explain what is depicted in the Table/Figure. E.g. Mean test accuracy, on 5 seeded trainings, with std.
Finally, details and references to e.g. appendix can be provided if necessary. E.g. Our method outperforms baseline 1 in 3 out of 4 tasks, ... etc.

Please keep the color attributed to each method consistent (i.e., use green for RESOLVE in *all* the Figures).

**Overclaims.** The authors sometimes overclaim. E.g. "We are the first to propose a strategy for addressing the relational bottleneck problem". What about this work: [1] ?

**Missing details on the evaluation.** The evaluation section needs a first paragraph that provides the reader with a lot of core details about the implementation, the metrics reported, the number of agent of each baseline and of the method, ... etc. I tend to think that the reported metrics are average (final?) training accuracies. Again, the captions need to be detailed further, as explained above. One scientific question should be answered with one Figure/Table. If more figures provide more insight, their should be placed in the appendix and referenced in the main text.

[1] Wüst, et al. "Pix2code: Learning to compose neural visual concepts as programs." UAI (2024).

**Questions:**

* How is it that transformers are overall better than abstractors in your experiments?
* Can you exhibit some relations in their vectorial forms ? Is their any compositionality? E.g. if you learn the sum and sign swap operations, can you get the subtraction one?
* Have you used different seeded training? Do you evaluate on training/testing/validation set?

---

### Official Review · Reviewer_M3kX · 2024-11-04

**Soundness:** 2
**Presentation:** 1
**Contribution:** 2
**Rating:** 3
**Confidence:** 3

**Summary:**

This work explores a novel vector symbolic architecture that allows superposition of relational representations and object-level features in high dimensional spaces. By experimenting on a series of benchmarks, RESOLVE demonstrates better performance than the state-of-the-art baselines.

**Strengths:**

**Originality: 4/5**

The concept of enabling both object-level features and relational representations to coexist within the same framework, without interference, is fascinating. Coupled with highly efficient operations at both the relational and feature levels, this novel vector symbolic architecture shows significant potential.

**Clarity: 1/5**

Pros: Figures 1 and 2 effectively illustrate the question domain and provide strong motivation. However,  in Figure 2, you have mentioned the quadratic equation solving task, while in the experimental section the task does not exist. This presentation is misleading.

Cons:

1. The methodology section is difficult to follow. Instead of presenting equations in structured blocks, the authors have embedded almost all mathematical expressions in lengthy paragraphs, making the content tedious and challenging to grasp. For example, instead of writing "R is normalized to obtain $\bar{R}$ using a Softmax function to produce probabilities," it would be clearer to simply write $\bar{R}$ = Softmax(R).
2. There are numerous comparisons between the architectures of baseline methods and the proposed approach. Attempting to focus on multiple elements simultaneously is distracting. Moving these comparisons to an appendix and concentrating on a detailed explanation of the proposed architecture would improve clarity.
3. The experimental setup is difficult to understand. Although multiple benchmarks are used, there is insufficient explanation for each. Clearly defining the input, output, and state-of-the-art (SOTA) baselines for each benchmark would help. Additionally, visualizations of each task would be very beneficial. Why are the baselines architecture-specific rather than task-specific? For example, what is the performance of GPT-4 with chain-of-thought (CoT) reasoning?

**Quality: 2/5**

The methodological details are intriguing, and the experiments yield promising results. However, the lack of clear presentation impedes confidence in the quality of the work.

**Significance: 3.5/5**

This vector symbolic architecture is likely to interest the neurosymbolic and vector-symbolic representation communities.

**Weaknesses:**

See strengths

**Questions:**

1. Why are the baselines architecture-specific rather than task-specific? For example, what is the performance of GPT-4 with chain-of-thought (CoT) reasoning?
2. What are clearly, the input and outputs of each tasks? SET task is clear, but what are even the objects in OBJECT-SORTING? In MATH PROBLEM-SOLVING, is the input Task name and question, and we expect the output as token sequence starting with "Answer"?

---

### Note · Authors · 2024-11-12

I have read and agree with the venue's withdrawal policy on behalf of myself and my co-authors.